# The Genomic and Genetic Evolution Analysis of Rabbit Astrovirus

**DOI:** 10.3390/vetsci9110603

**Published:** 2022-10-31

**Authors:** Qiaoya Zhao, Ye Tian, Liping Liu, Yifei Jiang, Haitao Sun, Shanjie Tan, Bing Huang

**Affiliations:** 1Institute of Poultry Sciences, Shandong Academy of Agricultural Sciences, Jinan 250100, China; 2Institute of Animal Husbandry and Veterinary Medicine, Shandong Academy of Agricultural Sciences, Jinan 250100, China; 3Linyi Animal Husbandry Development Promotion Center, Linyi 276600, China

**Keywords:** astrovirus, gene structure, gene evolution, virus isolation

## Abstract

**Simple Summary:**

Astrovirus is an emerging zoonotic pathogen. Rabbit astrovirus can cause diarrhea in rabbits, which brings great economic losses to the rabbit industry. In this study, a rabbit astrovirus strain named Z317 was successfully isolated from the carcasses of rabbits that suffered from diarrhea, and its whole genome sequence was obtained. In this study, we provide valuable data to help understand the variation and recombination of the virus, and lay a foundation for the prevention and control of the disease and molecular epidemiological research.

**Abstract:**

Rabbit astrovirus (RAstV) is a pathogen that causes diarrhea in rabbits, with high infection rate at various stages, which can often cause secondary or mixed infections with other pathogens, bringing great economic losses to the rabbit industry. In this study, 10 samples were collected from cases of rabbits with diarrhea on a rabbit meat farm in the Shandong area of China. The positive sample for astrovirus detected by RT-PCR was inoculated into an RK 13 cell line. A rabbit astrovirus strain named Z317 was successfully isolated, which produced an obvious cytopathic effect 48 h post-inoculation in the RK 13 cell line. The genome structure of this isolate was studied by high-throughput sequencing, showing that the Z317 strain had the highest similarity with the American strain TN/2208/2010, with 92.43% nucleotide homology, belonging to group MRAstV-23. The basic properties of the Z317 capsid (Cap) protein were analyzed, and 10 liner B cell epitopes were screened with the online biosoft Bepipred 2.0 and SVMTriP, including 445–464, 186–205, 655–674, 88–107, 792–811, 45–64, and 257–276 amino acids. This is the first contribution concerning RAstV genomes in China; more studies are needed to understand the diversity and impact of RAstV on rabbit health.

## 1. Introduction

Astroviruses (AstVs) belong to the independent Astroviridae family. According to their host types, they can be divided into two genera: mammalian astrovirus (*Mamastrovirius*) and avian astrovirus (*Avastrovirus*). They are a single-stranded, positive-strand RNA virus. Their genome size is about 6.4–7.7 kb, containing three open reading frames (ORF1a, ORF1b and ORF2) and polyadenylate tails. In 1975, AstV were first found in stool samples of infants hospitalized with diarrhea in Scotland by electron microscopy [1]. Subsequently, similar viruses were found in humans, livestock and wild animals [2], and bird [3] species, as well as in bats [4] and plants [5]. AstV infection is associated with gastroenteritis in most animals and humans. It has also been reported that AstVs are also related to parenteral diseases [6]. There is a high degree of variation in the genome of AstV. Gene recombination has been found in the ORF1b/ORF2 junction and ORF1b region of different AstV species [7], making astrovirus an emerging zoonotic pathogen. Cap protein can stimulate host cells to respond and participate in the assembly of virions, packaging nucleic acids, and virions maturation. Rabbit astrovirus was first found in the intestines and contents of Italian rabbits in 2011. It was detected in asymptomatic rabbits and diarrhea-suffering rabbits. The positive rate of symptomatic animals was 2–3 times higher than that of asymptomatic animals [8], and the positive rate of commercial rabbits was as high as 45% [9]. Due to the difficulty of in vitro isolation and infection testing, its pathogenicity is still unclear. In this study, astrovirus was detected in rabbit diarrhea samples by RT-PCR. Through virus isolation, virus macrogenome sequencing, genome sequence analysis, genetic evolution analysis, and Cap protein analysis, we hope to understand the variation and recombination of the virus, and lay a foundation for the prevention and control of the disease and molecular epidemiology research.

## 2. Materials and Methods

### 2.1. Materials

The intestinal tract and its contents were collected from 10 rabbits with clinical symptoms of diarrhea and then ground with saline at the rate of 1:3. After 3 repeated freeze–thaw cycles, the supernatant was centrifuged, and bacteria and tissue fragments in the samples were filtered out with a 0.22 μm filter to reduce the background of the samples.

### 2.2. RT-PCR

In total, 10 fecal samples were tested for common rabbit diarrheal pathogens by RT-PCR. Rabbit hemorrhagic disease virus (RHDV) [10], rotavirus [9], and coronavirus [11] primers were synthesized according to references. Based on existing astrovirus sequences (Genbank: JF729316.1), primers were designed with the Primer Premier 5.0 software (Primer Biosoft, Palo Alto, CA, USA) and synthesized by BGI gene Biotechnology Co., Ltd., and the primer sequences were as follows: rasvF—CTGGCTATACATCTGGCATAAC, rasvR—GCAGGCAGTCAACAAGTAG. An estimated 229 bp of cap gene fragment of astrovirus was amplified. Nucleic acid was extracted using the virus RNA/DNA nucleic acid extraction reagent from Tianlong Biology. The total RNA was stored at −80 °C until use. The reaction contained 2.0 μL of primers, 10.0 μL of 2× One-Step Mix, 0.5 μL of One-Step Enzyme Mix, 1.0 μL of extracted RNA, and double-distilled H_2_O diluted to a total volume of 20 μL. The RT-PCR program was as follows: a reverse transcription step at 50 °C for 30 min, initial denaturation at 95 °C for 5 min, and finally 35 amplification cycles at 95 °C for 20 s and 52 °C for 20 s, extension at 72 °C for 30 s, with a final extension for 10 min at 72 °C. The RT-PCR products were identified by gel electrophoresis. The PCR products were cloned into the pMD-18T vector (TaKaRa) and sequenced.

### 2.3. Virus Isolation

For virus isolation, a RK13 from Jiangsu Academy of Agricultural Sciences [12] was used. Approximately 500 μL of filter-sterilized supernatant from one sample of positive tissue fluid was diluted in the same volume of phosphate-buffered saline (PBS), and inoculated onto the RK13 monolayer, adding trypsin at a final concentration of 0.8 μg/mL. This was incubated at 37 °C for 2 h before the serum-free DMEM cell medium (GIBCO, Cat No.: C11995500BT) added. The inoculated cell cultures were further incubated further for 7 d at 37 °C in an incubator with 5% CO_2_ and were monitored daily for the presence of a cytopathogenic effect (CPE). The virus was harvested when the CPE of the inoculated cells reached 75%. The cells were inoculated and subcultured 6 times to observe cytopathic changes. Cell culture supernatants were collected for RT-PCR. The culture supernatant was stored at −80 °C.

### 2.4. Viral Genome Sequencing and Analysis

Nucleic acids from one astrovirus-positive materials were sent to Beijing Tsingke Biotechnology Co., Ltd. (Beijing, China) for library construction and high-throughput sequencing. High-throughput sequencing data were analyzed using a bioinformatics analysis platform. Using FastQC software (http://www.Bioinformatics.babraham.ac.uk/projects/fastqc/, accessed on 11 May 2020) to check sample data quality; this was follows by filtering low-quality repetitive sequences, and removing corresponding rRNA, host, and bacterial sequences by bbmap comparison with the NCBI nt database. The QC and the reads obtained after removing the contaminated sequence were assembled from scratch. SPAdes v3.11.1 (Bankevich, St. Petersburg, Russia) and MegaHit v1.1.2 (Dinghua Li, Tokyo, Japan) software were used to assemble the second-generation data. The contigs assembled were compared with the database, and the contigs with the highest similarity could be considered candidate reference genomes. The assembled scaffolds files were compared with the nt library to extract the virus sequences and thus obtain the virus annotation results.

Sequence analysis and phylogenetic tree construction of the existing rabbit astrovirus, representative strains of each species, and Cap protein sequences obtained by our laboratory were performed with MAGE 7 software (https://www.megasoftware.net accessed on 15 July 2020).

### 2.5. Bioinformatics Analysis of Cap Protein

The Cap protein’s basic physical and chemical properties were analyzed using the ProtParam online analysis tool (http://web.expasy.org/protparam/.net accessed on 21 July 2020). Using the DNAstar Protean program and Emimi, Kyte–Doolittle, and Jameson–Wolf methods, the surface probability plot, hydrophilicity plot, and antigenic index of the ORF2 coding capsid protein were predicted, respectively [13]. The transmembrane region was predicted online using the TMHMM software (https://www.cbs.dtu.dk/services/TMHMM/ accessed on 22 July 2020). The potential B cell linear epitopes of CAP protein were predicted and analyzed using the Bepipred 2.0 (http://www.cbs.dtu.dk/services/BepiPred/ accessed on 22 July 2020) and SVMTriP online analysis software (https://sysbio.unl.edu/SVMTriP/ accessed on 22 July 2020).

## 3. Results

### 3.1. RT-PCR Detection of RAstV

Out of 10 samples, 8 samples were positive according to RT-PCR, and sequencing confirmed that the sequence of cloned fragment was correct, with 6 samples positive for both astrovirus and rotavirus. RHDV and coronavirus were negative in all samples.

### 3.2. Virus Isolation

After inoculating the RK 13 cells with the samples positive for RT-PCR, a virus strain was isolated showing the characteristics of cell fusion, vacuoles on the surface of fused cells, round shrinkage, pulling nets, and finally falling off in sheets (Figure 1). One RAstV strain was successfully isolated, named Z317, and verified by RT-PCR.

### 3.3. Genome Composition

Metagenomic sequencing results show that probably no other viruses are associated with the disease in the samples from animals with diarrhea, except astrovirus. The whole genome sequence of astrovirus obtained by sequencing is 7,330 bp, and this has been uploaded to NCBI (GenBank: MZ682112). It contains ORF1a, ORF1b, ORF2, 3′UTR and part of the Poly (A) tail (Table 1).

In addition, conserved sequences such as the sliding sequence (AAAAAAC) exist in the genome, but not the conserved series of double-stem loop structures found in most astroviruses (Table 2).

### 3.4. Nucleotide Sequence Characteristics and Genetic Evolution Analysis

Nucleotide alignment was performed between the whole gene sequences of astrovirus found in our laboratory and those available in GenBank using MEGA 7.0 [17], and the similarity was found to be 90.85–93.44%. The amino acid sequence alignment showed the highest homology with TN/2208/2010 (NC_025346) (94.4%). The phylogenetic trees were constructed by MEGA 7.0 using the Maximum Likelihood Method (1000 bootstrap) for the ORF2 genes of each species, and the results show that Z317 was found in the same population as the American strain TN/2208/2010 (NC_025346) and the Italian strain Rabbit /Nausica/2008/ITA (JN052023) (Figure 2). Z317 belongs to MAstVAstV23, according to the latest genotypic classification criteria of astrovirus. Compared with other reference strains, ORF1b was the most conserved region, and ORF2 was the most mutated region. RDP analysis show that no recombination occurred in the Z317 strain.

### 3.5. Bioinformatics Analysis of Cap Protein

The Cap protein’s basic physical and chemical properties were analyzed by using the ProtParam online analysis tool (http://web.expasy.org/protparam/ accessed on 21 July 2020). The number of amino acids was 852, the molecular formula was C4063H6317N1165O1275S20, the relative molecular weight (MW) was 92,526.09, the theoretical isoelectric point (PI) was 6.06, and the pH was 7.0. It has a half-life of 30 h in mammals, more than 20 h in yeast, and more than 10 h in *Escherichia coli*. The instability coefficient is 36.61, and the hydrophilic coefficient (GRAVY) is −0.315. After the amino acid sequence of the Cap protein was uploaded to novopro, a nuclear localization signal was predicted, which was 9–58. Online prediction results using the TMHMM software (https://www.cbs.dtu.dk/services/TMHMM/ accessed on 22 July 2020) show that the protein had no transmembrane region.

Using the DNAstar Protean program and Emimi, Kyte–Doolittle, and Jameson–Wolf methods, a surface probability plot, hydrophilicity plot, and antigenic index of the ORF2 coding capsid protein were predicted, respectively [13]. The transmembrane region was predicted online using the TMHMM software. The analysis results show that the deduced encoded amino acid sequences were more likely on the protein surfaces of 1–150 aa 650–850 aa, and had similar antigenic indexes and hydrophilicity (Figure 3).

The potential B cell linear epitopes of the Cap protein were predicted and analyzed using the Bepipred 2.0 and SVMTriP online analysis software. The results show that the average antigen coefficient of the Cap protein predicted and analyzed by Bepipred 2.0 was 0.508. A total of 10 potential B cell epitopes were screened out (Table 3), among which 128–147 aa and 821–840 aa were the regions with the highest antigenicity, and thesewere located in the conserved region of astrovirus capsid protein, which could be used as the target region by the ELISA kit.

## 4. Discussion

To date, astroviruses have been detected in various birds and mammals. Astrovirus infection is species-specific, but in recent years, there have been more cross-species transmissions between wild animals and humans, wild animals and livestock and poultry, and increasing numbers are significantly different from the original classical strains at the genetic level. A new strain of astrovirus has been discovered, and studies have shown that genetic recombination occurs in humans [8], cattle, deer [18], turkey [19], and other animal astroviruses. The virus’ in vitro biological characteristics in cells can be used to simulate its in vivo replication and pathogenesis. Lee et al. found that human astrovirus could be serially passaged more than 13 times in human embryonic kidney cells (HEK) by adding trypsin [20]. Mi et al. isolated porcine astrovirus [21]. Goose astrovirus can be passaged by inoculating goose embryos [22]. So far, there is no cell culture for the isolation of rabbit astrovirus, and so the astrovirus is mostly detected by RT-PCR [9]. In this study, we successfully isolated an astrovirus strain named Z317, which was stable for more than six passages in a rabbit kidney cell line (RK 13) with trypsin (0.8 μg/mL final concentration). After 48h of treatment, cytopathic effects were observed, including syncytia formation, rounding of the cells, and vacuolization.

Many pathogens cause diarrhea in rabbits. In addition to pathogens such as *Escherichia coli* and *Salmonella*, RHDV, rotavirus, coronavirus, and astrovirus can all cause diarrhea. In this study, the positive rate of astrovirus was as high as 80% in 10 clinical samples with diarrhea, which means it may play a role in diarrhea or intestinal syndrome. Astrovirus-like particles have been found in rabbits with intestinal syndrome by electron microscopy [8]. Interestingly, the molecular detection of the positive samples of astrovirus showed that only two cases had no other common diarrhea pathogens, such as RHDV, rotavirus or coronavirus.. The results of metagenomic sequencing of the samples in this study show that no pathogens other than astroviruses could cause diarrhea, so it is inferred that astroviruses may play an important role in diarrhea in rabbits.

The classification of the genus astrovirus was originally proposed by the International Committee on Taxonomy of Viruses in 1995, from the initial species based on virion morphology to the origin of the virus, and finally, in 2010, the research group proposed a classification based on the amino acid sequence of the ORF2 genome [23]. The astrovirus ORF2 protein mainly determines the heterophilia of cells, and can stimulate host cells to respond to and participate in the assembly of virions, the packaging nucleic acids, and the maturation of virions [24]. From the sequence analysis of the full-length capsid sequence, it is worth noting that the 5′ end of ORF2 appears more conserved than the central region and the 3′ end. AstV contains a conserved region at the junction between ORF1b and ORF2; its exact role is unknown, but it may be a regulatory element of the subgenomic RNA encoding ORF2, a region that is almost completely conserved in this strain’s sequence. In addition, there is no stem-loop II-like motif (s2m) in this virus strain in the 3′ untranslated region of the genome [16]. This conserved sequence is present in most AstVs except for turkey AstV2, human AstV MLB1 [25], and rat AstV [26]. This sequence has been found in some coronaviruses and equine rhinitis viruses, and although its exact function is unknown, it may have broad functions in the biology of positive-strand RNA viruses [26]. A phylogenetic tree analysis showed that the rabbit astrovirus belongs to MAstV-23. The isolate presents high similarity (above 90%) to the nucleotide and aminoacid sequence with the ITA isolate and the US isolate. However, differences were larger for pathogenicity. The farm with the US strain showed mortality reaching as high as 90%, and in this study, the rate of mortality was less than 20%. However, the basic reason for this difference still needs further exploration.

In humans [27], ducks [28], geese [29], and pigs [30], there have been studies on protein screening, antigenicity analysis, and polyclonal antibody preparation for capsid proteins. Rabbit astrovirus fits in this category. The analysis results of this study show that the antigen index of most regions of the RAstV Cap protein is conserved, and is not much different from the American RAstV isolates published on the NCBI. It can thus be used as the target antigen for the development of new vaccines. Using bioinformatics software to predict and screen antigenic epitopes has many advantages, such as simplicity, rapidity, and low cost [31]. This study obtained 10 epitope peptides of the Cap protein of the RAstV Z317 strain through bioinformatics software analysis. However, whether it can induce good specific and non-specific immune responses in the body needs further verification. The next step will be related to virus isolation and animal experiments.

## 5. Conclusions

Here, we have isolated and fully described the genome of an astrovirus that is found in the feces of rabbits suffering from fatal intestinal diseases. Although the nucleic acid and amino acid sequences are highly similar to those previously reported, there are significant differences in mortality. The difference in pathogenicity needs further study. This study found that the infection rate of astrovirus in rabbits suffering from diarrhea was very high. Therefore, astrovirus should be added to the diagnosis and epidemic investigation of rabbit disease, so as to clarify the conditions of its transmission and incidence, and guide epidemic prevention practice.

## Figures and Tables

**Figure 1 vetsci-09-00603-f001:**
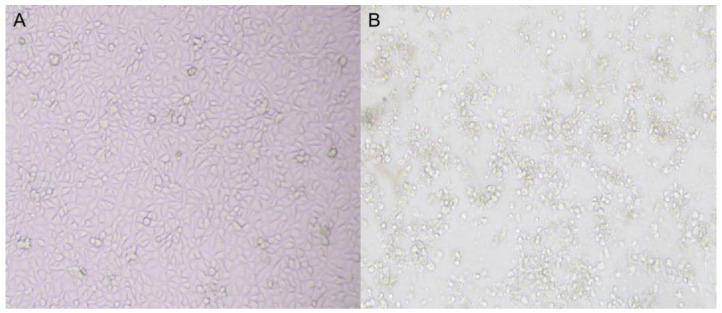
Lesions of Z317 isolate on RK 13 cells at 48 h (100×). (**A**): Normal RK 13 cell line. (**B**): Typical CPE of isolates in the RK 13 cell line.

**Figure 2 vetsci-09-00603-f002:**
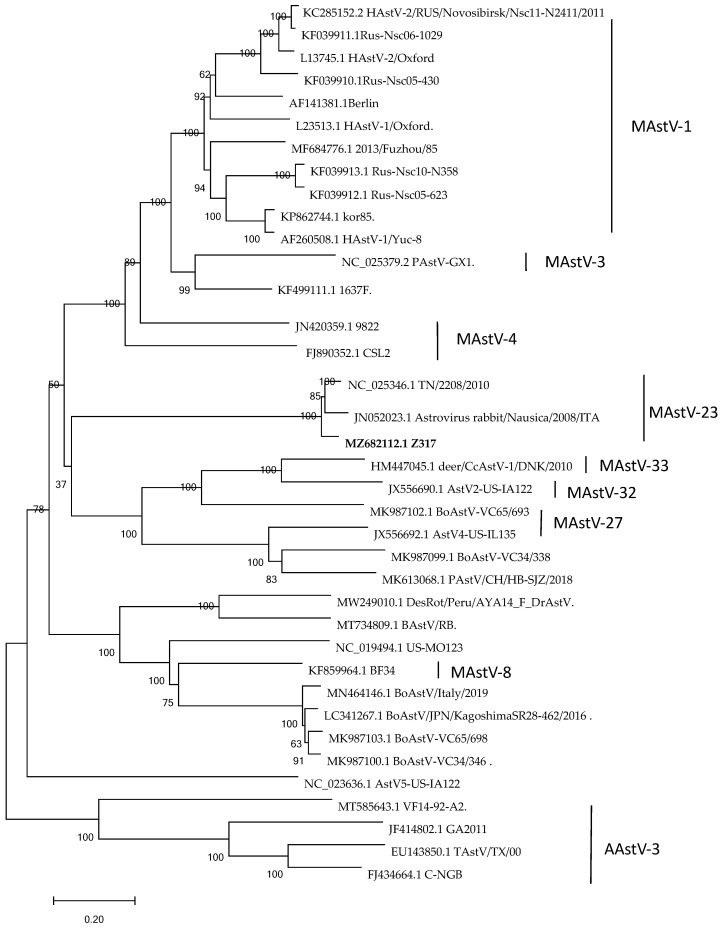
Genetic evolutionary tree of Cap gene of Z317 strain and reference strain. Reference strains from Genbank database. The phylogenetic tree was constructed by MEGA 7.0 using the Maximum Likelihood Method (1000 bootstrap). Bootstrap values > 60% were considered to be significant.

**Figure 3 vetsci-09-00603-f003:**
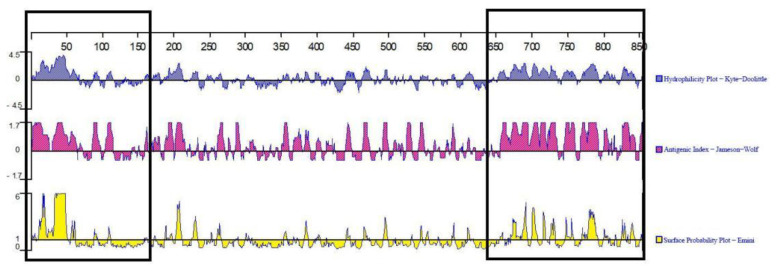
Analysis of physical and chemical properties of Cap protein. The hydrophilicity plot is shown in blue, the antigenic index is shown in red, the surface probability plot is shown in yellow, and the amino acids with a higher probability of deduced protein surfaces are circled in boxes.

**Table 1 vetsci-09-00603-t001:** Genome composition of the isolated strain.

Genome Component	Size	Deduced Amino Acid	Position
ORF1a	3201	1067aa	3–3203
ORF1b	1464	488aa	3224–4687
ORF2	2700	900aa	4539–7238
3′UTR	90	30aa	7239–7328

**Table 2 vetsci-09-00603-t002:** Conserved sequence characteristics of the isolated strain.

Name	Conserved Sequence Characteristics	Function	Position
Frameshift signal [14]	AAAAAAC	Induced ribosome translocation	2324–2330
Junction motif [15]	UUUGGAGNGGNGGACCNAAN4-8AUGNC	Subgenomic RNA transcription promoter	4656–4687
stem-loop Ⅱ motif [16]	stem-loop Ⅱ motif	Plays an important role in virus replication	-

**Table 3 vetsci-09-00603-t003:** Epitopes predicted inside the Cap protein.

Rank	Location	Epitope	Score
1	128–147	RCVQAHIRFTPLVGSSAVSG	1.000
2	821–840	VSRFDYNRVERGMSNLEAKK	0.878
3	308–327	TGTSVSSTIFQVVDASVSTA	0.738
4	445–464	LNGWIQNHVNAVVGLWIQDS	0.614
5	186–205	LARKQLAGPRESWWLTNTND	0.594
6	655–674	EWQRASGARVDLRTVRFRDD	0.578
7	88–107	SSDRVELEMASMLNPALVKE	0.513
8	792–811	QSALGAQFDPETAAHRAMRA	0.471
9	45–64	QQRRTRAVARSEVKREVHRL	0.416
10	257–276	ATLSKQEAPASSVVIDAATA	0.402

## Data Availability

The datasets generated and/or analyzed in the current study are available in the GenBank database and in the online Appendix A.

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
