# Peer review of "The Genomic and Genetic Evolution Analysis of Rabbit Astrovirus"

_vetsci, 2022, doi:10.3390/vetsci9110603_

Round 1

Reviewer 1 Report

Comments and suggestions for Authors

The submitted manuscript entitled „The Genomic and Genetic Evolution Analysis of Rabbit Astrovirus” described a successfully isolated astrovirus strain from carcasses of rabbits suffered from diarrhea. This is a valuable manuscript because this virus is a potential zoonotic pathogen.

Article is well constructed, materials and methods are well described and developed. Results are clearly presented. However, there are many stylistic and punctuation errors and I suggest that it need some minor corrections and text editing before publication. Furthermore, there is lack of summary/conclusion of the manuscript. Please find the rest of comments and suggestions id the pdf file.

Author Response

Re: Submission ID vetsci-1956853 (The Genomic and Genetic Evolution Analysis of Rabbit Astrovirus)

Dear Editor,

On behalf of all co-authors, I would like to thank you and the reviewers for the very

informative and constructive comments and suggestions for our manuscript The Genomic and Genetic Evolution Analysis of Rabbit Astrovirus). We have taken all comments/suggestions into consideration in the revised manuscript and provided a detailed point-by-point response to your specific comments in the attached list below. Also our manuscript has been edited for English language usage, grammar, spelling and punctuation by native speakers at https://www.mdpi.com/authors/english (english-52143).  

We sincerely hope that the revision will make it more suitable for “Veterinary Sciences”. We deeply appreciate your consideration of our manuscript. If you and the reviewers have any

other questions, please feel free to let us know.

Thank you and best regards.

Sincerely,

Qiaoya Zhao

Institute of Poultry Sciences, Shandong Academy of Agricultural Sciences

*Corresponding author: Bing Huang

address: Institute of Poultry Sciences, Shandong Academy of Agricultural Sciences, Jinan,

Shandong Province 250023, China. E-mail: hbind@163.com

List of point-by-point response to the Editor's comments

Comments from the Review Editors:

Reviewer 1

  1. However, there are many stylistic and punctuation errors and I suggest that it need some minor corrections and text editing before publication.

Response: Thanks very much for your careful review. In the revised manuscript, we have checked the manuscript carefully including the format and style, and all the modified portions are highlighted in red. Also our manuscript has been edited for English language usage, grammar, spelling and punctuation by native speakers at https://www.mdpi.com/authors/english.

  1. Furthermore, there is lack of summary/ of the manuscript.

Response: Thanks for the suggestion. We have added conclusion in the revised manuscript. 

  1. Please find the rest of comments and suggestions inthe pdf file.

Response: Thanks for your kindly suggestion. The text of the manuscript has been revised according to the comments and suggestions.

Reviewer 2 Report

In the manuscript entitled “The Genomic and Genetic Evolution Analysis of Rabbit Astrovirus” authors have done genomic characterization of Rabbit Astrovirus from the samples collected from a rabbit farm. Authors ruled out other related infection by RT-PCR and isolated RAstV in RK-13 cells which was evident by the cytopathic effects and verified by RT-PCR. Next generation sequencing of a positive clinical sample was done to obtain the complete genome sequence of the RAstV.

I have a few concerns which are as follows:

1.     Authors need to do extensive language corrections throughout the manuscript. I suggest authors to take help from a native English speaker or use English language editing service to improve the language of this manuscript.

2.     In the simple summary (Line 12) authors have mentioned that they successfully isolated a rabbit rotavirus strain named Z317.

3.     In the section 3.1: Authors have mentioned that they ligated the amplified cap gene product in the cloning vector by double enzyme digestion though the sequence of the cap gene provided by the authors don’t have any restriction site. Authors should clarify this.

4.     Authors have also mentioned that they identified recombinant plasmid by PCR but they didn’t mention that from where they got these recombinant plasmids.

5.     Authors should upload the raw metagenomics data files on public portal for the verification of the results.

6.     In Figure 2 legends: Authors mentioned they marked their sequence with purple background. No such marking was found in the figure.

7.     Line 213: Correct typo error AstV written two times.

8.     Line 226 and 227: Authors have written RAstV AstV

9.     References need to be formatted as per journal’s policy. I recommend authors to use the citation tool (eg. Mendeley, Endnote, etc.) to meet the journal’s guidelines to properly cite the references.

Author Response

Re: Submission ID vetsci-1956853 (The Genomic and Genetic Evolution Analysis of Rabbit Astrovirus)

Dear Editor,

On behalf of all co-authors, I would like to thank you and the reviewers for the very

informative and constructive comments and suggestions for our manuscript The Genomic and Genetic Evolution Analysis of Rabbit Astrovirus). We have taken all comments/suggestions into consideration in the revised manuscript and provided a detailed point-by-point response to your specific comments in the attached list below. Also our manuscript has been edited for English language usage, grammar, spelling and punctuation by native speakers at https://www.mdpi.com/authors/english (english-52143).  

We sincerely hope that the revision will make it more suitable for “Veterinary Sciences”. We deeply appreciate your consideration of our manuscript. If you and the reviewers have any

other questions, please feel free to let us know.

Thank you and best regards.

Sincerely,

Qiaoya Zhao

Institute of Poultry Sciences, Shandong Academy of Agricultural Sciences

*Corresponding author: Bing Huang

address: Institute of Poultry Sciences, Shandong Academy of Agricultural Sciences, Jinan,

Shandong Province 250023, China. E-mail: hbind@163.com

List of point-by-point response to the Editor's comments

Comments from the Review Editors:

  1. Authors need to do extensive language corrections throughout the manuscript. I suggest authors to take help from a native English speaker or use English language editing service to improve the language of this manuscript.

Response: Thanks very much for your careful review. Our manuscript has been edited for English language usage, grammar, spelling and punctuation by native speakers at https://www.mdpi.com/authors/english. 

  1. In the simple summary (Line 12) authors have mentioned that they successfully isolated a rabbit rotavirus strain named Z317.

Response: Thanks for your kindly suggestion. This was a mistake, thanks for spotting it. Now corrected.

  1. In the section 3.1: Authors have mentioned that they ligated the amplified cap gene product in the cloning vector by double enzyme digestion though the sequence of the cap gene provided by the authors don’t have any restriction site. Authors should clarify this.

Response: Thanks very much for your careful review. We did use enzyme digestion to identify plasmids when establishing detection methods, double enzyme digestion was not used in the identification here. We didn't describe it clearly enough to create ambiguity. This is my misstake. Thank you for your comments and help us correct this.

  1. Authors have also mentioned that they identified recombinant plasmid by PCR but they didn’t mention that from where they got these recombinant plasmids.

Response: Thanks very much for your careful review. This have been explained in the methods section.

  1. Authors should upload the raw metagenomics data files on public portal for the verification of the results.

Response: Thanks for your kindly suggestion. Some research on the original data is still in progress, and relevant articles have not been published, so we will disclose the original data soon. We have uploaded the sequence of rabbit astrovirus in the data to the public platform (https://www.ncbi.nlm.nih.gov/) and got the serial number (Genbank: MZ682112).

  1. In Figure 2 legends: Authors mentioned they marked their sequence with purple background. No such marking was found in the figure.

Response: Thanks very much for your careful review. This is a mistake. Our sequence have been marked in boldface.

  1. Line 213: Correct typo error AstV written two times.

Response: Thanks very much for your careful review. Modified.

  1. Line 226 and 227: Authors have written RAstV AstV

Response: Thanks very much for your careful review. Modified.

  1. References need to be formatted as per journal’s policy. I recommend authors to use the citation tool (eg. Mendeley, Endnote, etc.) tomeet the journal’s guidelines to properly cite the references.

Response: Thanks very much for your careful review. Modified.

Reviewer 3 Report

Dear authors!

Thank you for preparing such an interesting paper on Rabbit Astroviruses. I have some minor comments to improve your article (in appendix).

For Introduction I suggest you add description on genome organization, data according to classification and description (functions) of CAP.

In Materials and Methods please add precise description on sample size (how many samples analysed by each method) and description of protocols used. I would also suggest adding references and versions of software. Please also add  description of plasmid production and PCR. It is also very important to add description of how phylogenetic tree was constructed (method and references to the models used).

In Results section I would suggest adding a Table with listed sequences (with host, year of isolation and location data) that were used for phylogenetic analysis. Also add references and versions of software.

In Discussion I suggest to add comparison of your valuable results (sequence) to published studies.

Thank you and best wishes!

Author Response

Re: Submission ID vetsci-1956853 (The Genomic and Genetic Evolution Analysis of Rabbit Astrovirus)

Dear Editor,

On behalf of all co-authors, I would like to thank you and the reviewers for the very

informative and constructive comments and suggestions for our manuscript The Genomic and Genetic Evolution Analysis of Rabbit Astrovirus). We have taken all comments/suggestions into consideration in the revised manuscript and provided a detailed point-by-point response to your specific comments in the attached list below. Also our manuscript has been edited for English language usage, grammar, spelling and punctuation by native speakers at https://www.mdpi.com/authors/english (english-52143).  

We sincerely hope that the revision will make it more suitable for “Veterinary Sciences”. We deeply appreciate your consideration of our manuscript. If you and the reviewers have any

other questions, please feel free to let us know.

Thank you and best regards.

Sincerely,

Qiaoya Zhao

Institute of Poultry Sciences, Shandong Academy of Agricultural Sciences

*Corresponding author: Bing Huang

address: Institute of Poultry Sciences, Shandong Academy of Agricultural Sciences, Jinan,

Shandong Province 250023, China. E-mail: hbind@163.com

  1. For Introduction I suggest you add description on genome organization, data according to classification and description (functions) of CAP.

Response: Thanks very much for your careful review. I had added description on genome organization, data according to classification and the function of cap protein in the introduction.

  1. In Materials and Methods please add precise description on sample size (how many samples analysed by each method) and description of protocols used. I would also suggest adding references and versions of software. Please also add description of plasmid production and PCR. It is also very important to add description of how phylogenetic tree was constructed (method and references to the models used).

Response: Thanks for your kindly suggestion. we have extensively modified in the text. and all the modified portions are highlighted in red.

  1. In Results section I would suggest adding a Table with listed sequences (with host, year of isolation and location data) that were used for phylogenetic analysis. Also add references and versions of software.

Response: Thanks for your kindly suggestion. The data have been provided as Supplementary table. 1 in the Supplementary Information. References and versions of software have been added.

  1. In Discussion I suggest to add comparison of your valuable results (sequence) to published studies.

Response: Thanks for your kindly suggestion. I have added comparison of all my data to published studies on astroviruses in rabbits - include the closest sequences from Italy and USA.

Round 2

Reviewer 2 Report

This revised mauscript is in a better shape to be accepted.

I need a small clarification from the authors in Line 91 where they mentioned that "This was incubated at 37 °C for 2 h before the serum-free DMEM cell medium" whether the authors added serum-free DMEM or full growth media to grow the infected cells.

If its just a typo error, it should get corrected before its acceptance.

Author Response

Re: Submission ID vetsci-1956853 (The Genomic and Genetic Evolution Analysis of Rabbit Astrovirus)

Dear Editor,

On behalf of all co-authors, I would like to thank you and the reviewers for the very

informative and constructive comments and suggestions for our manuscript The Genomic and Genetic Evolution Analysis of Rabbit Astrovirus). We have taken all comments/suggestions into consideration in the revised manuscript and provided a detailed point-by-point response to your specific comments in the attached list below. 

  1. I need a small clarification from the authors in Line 91 where they mentioned that "This was incubated at 37°C for 2 hbefore the serum-free DMEM cell medium" whether the authors added serum-free DMEM or full growth media to grow the infected cells. If its just a typo error, it should get corrected before its acceptance.

Response: Thanks very much for your careful review. Its not a typo error, trypsin was used in the incubation stage, the serum-free DMEM was indeed added to grow the infected cells.

We sincerely hope that the revision will make it more suitable for “Veterinary Sciences”. We deeply appreciate your consideration of our manuscript. If you and the reviewers have any

other questions, please feel free to let us know.

Thank you and best regards.

Sincerely,

Qiaoya Zhao

Institute of Poultry Sciences, Shandong Academy of Agricultural Sciences

*Corresponding author: Bing Huang

address: Institute of Poultry Sciences, Shandong Academy of Agricultural Sciences, Jinan,

Shandong Province 250023, China. E-mail: hbind@163.com